# Uniform Complexity for Text Generation

**Joseph Marvin Imperial**[Ω,Λ]   **Harish Tayyar Madabushi**[Λ]
[Λ]University of Bath, UK
[Ω]National University, Philippines
jmri20@bath.ac.uk  htm43@bath.ac.uk

## Abstract

Large language models (LLMs) have shown promising results in a wide array of generative NLP tasks, such as summarization and machine translation. In the context of narrative generation, however, existing models still do not capture factors that contribute to producing consistent text. For instance, it is logical that a piece of text or a story should be *uniformly readable* throughout and that this form of complexity should be controllable. As such, if the complexity of an input text prompt is rated first-grade reading level in the Flesch Reading Ease test, then the generated text continuing the plot should also be within this range of complexity. With this in mind, we introduce **Uniform Complexity for Text Generation (UCTG)**, a new benchmark test which raises the challenge of making generative models observe uniform linguistic properties with respect to prompts. We experiment with over 150+ linguistically and cognitively motivated features for evaluating text complexity in humans and generative models. From our results, we find that models such as GPT-2 struggle to preserve the complexity of input prompts used in its generations, even if finetuned with professionally written texts[1].

## 1 Introduction

Consistency is key in all aspects of the story-writing process. A story has to have a consistent plot or story arc, observe a consistent use of factual information if needed, make use of consistent personas for characters, and follow a consistent writing style in order to be comprehensible (Guan and Huang, 2020; Zhang et al., 2022; Huang et al., 2023). While topic consistency is often used as a general metric for evaluating the quality of texts produced by narrative generation systems (Roemmele, 2021), an important aspect that has received less attention from the research community is the

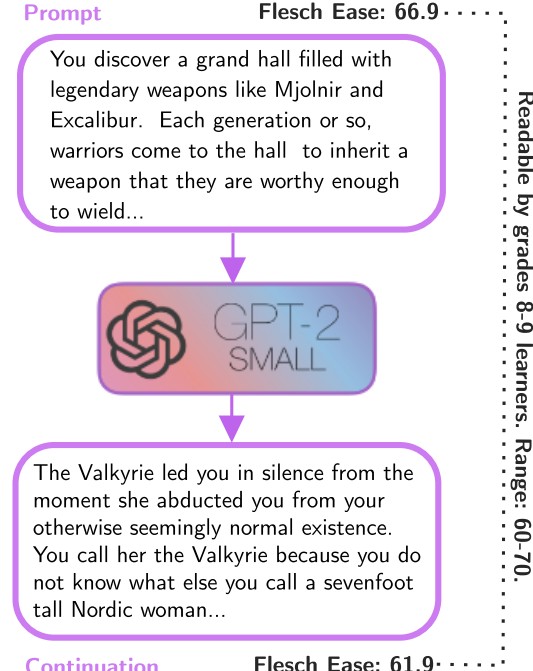

Prompt          Flesch Ease: 66.9

You discover a grand hall filled with legendary weapons like Mjolnir and Excalibur. Each generation or so, warriors come to the hall to inherit a weapon that they are worthy enough to wield...

The Valkyrie led you in silence from the moment she abducted you from your otherwise seemingly normal existence. You call her the Valkyrie because you do not know what else you call a sevenfoot tall Nordic woman...

Continuation          Flesch Ease: 61.9

Readable by grades 8-9 learners. Range: 60-70.

Figure 1: An illustrated ideal example of a generative model (e.g. GPT-2) producing **uniformly complex** text where the Flesch Ease score of the generated text falls within the same readability range as the prompt which is 60-70 or 8th-9th grade reading level.

investigation of consistency of readability or *complexity* of texts.

When crafting a story, it is very reasonable for a writer to ensure that all sentences are readable by a specific target audience (i.e., first-grade learners). As such, one must take note of common words within the general vocabulary of readers belonging to a specific grade to avoid possible frustration in reading that will hinder effective comprehension (Gickling and Armstrong, 1978; Guevarra, 2011). Thus, a text's complexity (and so the readability) depends largely on the writer's capability to ensure that the complexity of words and sentences is consistent (Fountas and Pinnell, 1999; DuBay, 2004;

---

[1]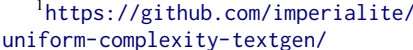https://github.com/imperialite/
uniform-complexity-textgen/

Agrawal and Carpuat, 2019). In the text generation process, since the output of generative models is controlled by the prompt, it is essential that the complexity of the output to be consistent with the input prompt. Figure 1 shows an ideal scenario where a prompt given by a human has the same level of reading difficulty (66.9) as the generated text by a GPT-2 model (61.9) based on the Flesch Ease formula (Flesch, 1948). The interpretation of Flesch Ease scores obtained means both texts can be read easily by Grade 8 and 9 learners.

Generally, the complexity of a narrative or any piece of text at hand can be measured by a multitude of content-based and linguistic factors. Some measures that have been explored through the years include **syntactic complexity** as equivalence of high-proficiency writing through the presence of qualified syntactic phrases or chunks such as words from a noun or verb phrase (Beers and Nagy, 2009; McNamara et al., 2010; Roemmele et al., 2017), **discourse complexity** by aggregating the presence of entities of a text such as mentions people, organizations, and locations which can lead to increased working memory burden (Feng et al., 2009), and **vocabulary complexity** by matching words from the text associated with a specific age-based difficulty level from developmental studies (Kuperman et al., 2012; Vajjala and Meurers, 2016) to name a few. These factors, when extracted from texts for evaluating complexity, usually follow a general linear property where the complexity of a text also increases as the value of a measured feature is increased (Genzel and Charniak, 2002).

In this study, we aim to answer the following research question: **can large language models (LLMs) generate responses that observe uniform complexity with the prompt?** To answer this, we introduce the novel task of Uniform Complexity for Text Generation (UCTG) with the aim of investigating responses obtained from any arbitrary generative language models used in story generation With GPT-2 models as test case, we explore how the outputs generated by LLMs relate to prompts and continuations given by humans, in relation to these linguistic features. Specifically, we make the following major contributions:

1. We empirically show the inconsistencies in linguistic characteristics exhibited by LLMs, specifically GPT-2, with respect to the complexity of the prompts used. We cover a wide array of linguistic complexity features (total

160) used for evaluating text complexity in this process. We also report that fine-tuned models, even trained on uniformly complex and professionally written text, do not alleviate this problem.

2. Despite inconsistencies with respect to the prompts, we show that there are some similarities with the way GPT-2 models (fine-tuned or not) and humans generate continuations, such as the preference of words from pre-defined word familiarity vocabularies such as Age-of-Acquisition and SubtlexUS.

3. Scoping our narrative more broadly, we sketch possible careful considerations when framing UCTG as an automatic evaluation metric for NLG systems due to the scalability of text complexity across language-related tasks.

## 2 Benchmark Task Description

The proposed evaluation task can be generally adapted to any NLG framework as long as texts are both the input and output. In the case of a narrative generation setting, given a series of prompts where each one is made up of a few sentences $P = \{p_1, p_2, \ldots, p_n\}$ and their continuations $C = \{c_1, c_2, \ldots, c_n\}$ by a model $M$, a linguistic feature function $F$ processes instances from both the prompt and continuation sets to get their corresponding measures $F(p_n) \in \mathbb{R}$ and $F(c_n) \in \mathbb{R}$. All calculations performed by $F$ are normalized[2] with either the number of word counts or a similar variable to avoid higher values from longer generated texts. A statistical test is then applied using the two groups to measure the significant differences in each linguistic complexity feature variable.

## 3 Generation Setup

We describe the training recipes and resources used for exploring UCTG in human and GPT-2 models.

### 3.1 Human Prompt and Continuation Data

For the compilation of prompts and human continuation as a benchmark, we used the WRITING-PROMPTS[3] dataset collected by Fan et al. (2018). This dataset is derived from the r/WritingPrompts community of the Reddit platform where it allows

---

[2]This is a common practice in text complexity research to mitigate the effects of length in feature values (Lu, 2012).
[3]www.reddit.com/r/WritingPrompts/

users to post story premises in various contexts and genres for other members to *continue* with their own ideas and creativity. The current compilation is divided into train, test, and validation splits. Given the nature of this task, we only used the test split which contains 15,138 pairs. In addition to the data preprocessing steps done by DeLucia et al. (2021), we handpicked prompts with at least 30 words in length to ensure that the neural models will have enough context for the generation phase while limiting human continuation texts to a minimum of 150 words and a maximum of 300 words to avoid long-range memory limitation in the small GPT-2 model used (Sun et al., 2021). Overall, we arrived at a total of 941 prompt-human continuation pairs as the benchmark for the task.

## 3.2 Generation Models

To test whether large neural language models are capable of UCTG, we selected the GPT-2 class of models (Radford et al., 2019) for analysis due to its notable extensive use in the NLP community for the narrative generation task (See et al., 2019; Xu et al., 2020; Akoury et al., 2020; Chang et al., 2021). Aside from human generations at hand, we used three variations of GPT-2[4] for comparison as listed below:

**Base GPT-2**. This model is the standard off-the-shelf medium version with 355M parameters from Huggingface. We refer to this model as the base version as it is used without any fine-tuning.

**Finetuned GPT-2 for General Narrative Generation** (DeLucia et al., 2021). This model uses the GPT-2 medium version that has been fine-tuned for general narrative generation using the same WRITINGPROMPTS dataset in Fan et al. (2018) and has been optimized with the best hyperparameters for maximum fluency and diversity of content.

**Finetuned GPT-2 with Uniformly Complex Texts**. This model also uses the GPT-2 medium version but we fine-tuned it using datasets from NEWSELA and ELG. NEWSELA[5] is a collection of 10,787 professionally-written news articles in English with over 4-5 different variations in reading levels (Scarton et al., 2018; Xu et al., 2016, 2015). We also add another data source from

the European Language Grid (ELG)[6] compiled by Breuker (2022) which contains 1,200 curated stories in all levels of the Common European Framework of Reference for Languages (CEFR). We refer to both of these language resources as *professionally-written* and *uniformly complex* as they have been produced and classified by expert writers in contrast to the WRITINGPROMPTS dataset in Fan et al. (2018) where the authors of narrative continuations are Reddit users and are non-experts. For the fine-tuning setup, we use the same recipe found in DeLucia et al. (2021). We upload the finetuned model in Hugginface for public access[7].

For the decoding setup, DeLucia et al. (2021) reports that the best values for nucleus sampling or top-$p$ for the narrative generation task range from 0.70 to 0.90 wherein generated stories are more vivid and of better quality as evaluated through human and automatic means. Thus, for this work, we set top-$p$ to 0.90 for all GPT-2 models.

## 4 Testing for (In)consistencies from the Prompt-Continuation Pairs

For the first experiment, we test for possible significant differences in the complexity values of the prompts and generations of the three variations of GPT-2 models described previously plus generations from humans. We use the **Welch t-test** (Welch, 1947) with Bonferroni correction resulting in a new alpha level of $\alpha = 0.0125$ (0.05/4). We extracted 160 linguistic features using the **LingFeat tool** by Lee et al. (2021) for the variables of interest. The tool covers the extraction of surface, lexical, phrasal, tree structure, type token, psycholinguistics, and formula-based readability features as further detailed in the subsections below. We only used the normalized or average-based linguistic complexity features (ex. count of noun POS / total count of words) instead of total raw counts to avoid bias or reaching extremely high values for longer sentences. As a reference for labels of models used in the succeeding tables, we use **GPT2** for the base model with no fine-tuning, **GPT2-DF** for the model fine-tuned for general narrative generation, and **GPT2-UF** for the model fine-tuned with uniformly complex dataset.

---

[4]https://huggingface.co/gpt2
[5]https://newsela.com/data/

[6]https://live.european-language-grid.eu/catalogue/corpus/9477
[7]https://huggingface.co/josephimperial/uniform-complex-gpt2-med

For ease of readability and display purposes, we report the *p*-values from the result of the Welch test for the tables per linguistic group in the succeeding sections. We also include an extensive set of barplot visualizations in Appendix A arranged alphabetically for ease of referencing.

## 4.1 Shallow Features

For our first feature set, we looked at 8 shallow or surface-based textual features as shown in Table 1 such as *product and square root of total tokens by total sentences, log densities of tokens to sentences, average token count per sentence, average syllable count per sentence and per token, and average character count per sentence and token*. These linguistic features have been extensively used text complexity assessment in a wide range of languages such as in English (Flesch, 1948), French (François and Miltsakaki, 2012) and Filipino (Imperial and Ong, 2021). From the result, all of the mentioned features for the four groups appear to be significantly different with respect to the complexity of the prompt except for *average characters per token* for the finetuned GPT-2 model with uniformly complex data. From this, we posit that both humans and GPT-2 models, regardless of applied fine-tuning, *generally* ignore shallow or surface-based characteristics of texts with respect to the prompt.

| Feature | Human | GPT2 | GPT2-DF | GPT2-UF |
|---|---|---|---|---|
| Total token x Total sent | 0.0000 | 0.0000 | 0.0000 | 0.0000 |
| Sqrt Total token x Total sent | 0.0000 | 0.0000 | 0.0000 | 0.0000 |
| Log token / Log sent | 0.0000 | 0.0000 | 0.0000 | 0.0000 |
| Avr token sent | 0.0000 | 0.0000 | 0.0000 | 0.0000 |
| Avr Syll sent | 0.0000 | 0.0000 | 0.0000 | 0.0000 |
| Avr Syll token | 0.0000 | 0.0000 | 0.0000 | 0.0000 |
| Avr Chars sent | 0.0000 | 0.0000 | 0.0000 | 0.0000 |
| Avr Chars token | 0.0020 | 0.0000 | 0.0000 | **0.1281** |

Table 1: Shallow based features.

## 4.2 Traditional Formula-Based Features

Formula-based features for readability assessment also stem from a combination of surface-based features such as word length, sentence length, and occurrence from a pre-defined dictionary of words. We covered 6 metrics such as *Flesch-Kincaid* (Kincaid et al., 1975), *New Automated Readability Index (NARI)* (Smith and Senter, 1967), *Coleman-Liau* (Coleman and Liau, 1975), *SMOG* (Mc Laughlin, 1969), *Gunning-Fog* (Gunning et al., 1952), and *Linsear* (Klare, 1974). Due to their ease of use, formulas such as the Flesch-Kincaid Reading

Ease are integrated into most text-based applications. From the results in Table 2, the majority of the narrative continuations from humans, base GPT-2 model, and both the finetuned GPT-2 models were significantly different except for two. The formula for NARI is the only one that uses the number of characters as a predictor, which may signal that human-continued texts are more sensitive to character count. In addition, the base GPT-2 model obtaining non-significance for text dissimilarity on Flesch-Kincaid may indicate potential preservation of complexity. However, this is a one-off and is not consistent for all formulas explored.

| Feature | Human | GPT2 | GPT2-DF | GPT2-UF |
|---|---|---|---|---|
| Flesch-Kincaid | 0.0008 | **0.0831** | 0.0000 | 0.0000 |
| NARI | **0.3535** | 0.0111 | 0.0000 | 0.0000 |
| Coleman-Liau | 0.0000 | 0.0000 | 0.0000 | 0.0000 |
| SMOG | 0.0000 | 0.0000 | 0.0026 | 0.0000 |
| Gunning-Fog | 0.0000 | 0.0002 | 0.0000 | 0.0000 |
| Linsear | 0.0000 | 0.0005 | 0.0000 | 0.0000 |

Table 2: Traditional formula-based features.

## 4.3 Part of Speech Features

We further the analysis by looking deeper into the text structure via part-of-speech (POS) tags. For this study, syntactic concepts covering *nouns, verbs, adverbs, adjectives, subordinating clauses and conjunction, coordinating clauses and conjunction* and *prepositional phrases* were used as predictors to calculate the densities of prompts and text continuations in both sentence and token level aspect (Heilman et al., 2007; Lu, 2012). Table 3 details 47 ratio and average-based predictors, quantifying POS complexities of human and neural model text continuations. Overall, the finetuned GPT-2 model for general story generation obtained the least number of complexity features that are significantly different from the prompt with 31 (16 features non-significant). This is followed by the human-generated continuations with 33 (14 features non-significant), the baseline GPT-2 with 37 (10 features non-significant), and GPT-2 model trained from uniformly complex data coming in last with 41 (6 features non-significant). One clear observation that can be derived from the POS features is that all non-significance from the finetuned GPT-2 model for uniformly complex data overlaps with the continuations from humans and the finetuned GPT-2 model for general story generation, especially for features related to the average use of *adjective words, content words*, and *coordi-*

*nating conjunctions*. We infer that the additional fine-tuning process done for neural models encourages some level of uniformity of usage of the said grammatical entities with respect to prompts. Interestingly, fine-tuning on a general story completion dataset encourages more sensitivity to these features compared to using uniformly complex text. This is surprising as one might expect increased consistency in the generated output when trained on such data. This result further highlights the need for the evaluation of complexity in language model outputs and for this task, which is non-trivial to solve.

| Feature | Human | GPT2 | GPT2-DF | GPT2-UF |
|---|---|---|---|---|
| Avr Noun POS sent | 0.0000 | **0.0194** | 0.0000 | 0.0000 |
| Avr Noun POS token | 0.0000 | 0.0000 | 0.0000 | 0.0000 |
| Noun POS to Adj POS | 0.0002 | **0.1467** | 0.0025 | 0.0000 |
| Noun POS to Verb POS | 0.0000 | 0.0000 | 0.0000 | 0.0000 |
| Noun POS to Advrb POS | 0.0000 | 0.0000 | **0.3215** | 0.0000 |
| Noun POS to SubrdConj | 0.0000 | 0.0000 | 0.0096 | 0.0000 |
| Noun POS to CordConj | 0.0000 | **0.0814** | 0.0002 | 0.0118 |
| Avr Verb POS sent | **0.1531** | 0.0000 | 0.0000 | 0.0000 |
| Avr Verb POS token | 0.0000 | **0.6279** | 0.0000 | 0.0000 |
| Verb POS to Adj POS | **0.0244** | 0.0006 | 0.0000 | 0.0006 |
| Verb POS to Noun POS | 0.0000 | 0.0000 | 0.0000 | 0.0000 |
| Verb POS to Advrb POS | **0.2302** | 0.0000 | 0.0000 | 0.0123 |
| Verb POS to SubrdConj | 0.0000 | 0.0000 | 0.0000 | 0.0000 |
| Verb POS to CordConj | 0.0000 | 0.0000 | 0.0000 | 0.0000 |
| Avr Adj POS sent | **0.0557** | **0.0735** | 0.0000 | 0.0000 |
| Avr Adj POS token | **0.7609** | 0.0013 | **0.6105** | **0.0249** |
| Adj POS to Noun POS | **0.1982** | 0.0027 | 0.0000 | 0.0000 |
| Adj POS to Verb POS | 0.0000 | 0.0092 | 0.0024 | 0.0000 |
| Adj POS to Advrb POS | 0.0000 | 0.0000 | **0.0506** | 0.0000 |
| Adj POS to SubrdConj | 0.0000 | 0.0000 | **0.0408** | 0.0000 |
| Adj POS to CordConj | 0.0000 | 0.0011 | **0.5120** | **0.6421** |
| Avr Advrb POS sent | 0.0000 | **0.0411** | 0.0000 | 0.0003 |
| Avr Advrb POS token | 0.0000 | **0.0668** | **0.3566** | 0.0000 |
| Advrb POS to Adj POS | 0.0000 | **0.0726** | **0.0429** | 0.0000 |
| Advrb POS to Noun POS | 0.0000 | 0.0008 | 0.0000 | 0.0000 |
| Advrb POS to Verb POS | 0.0000 | 0.0047 | 0.0000 | 0.0000 |
| Advrb POS to SubrdConj | 0.0000 | 0.0000 | 0.0138 | 0.0000 |
| Advrb POS to CordCobj | 0.0000 | **0.0247** | **0.5228** | 0.0000 |
| Avr SubrdConj sent | **0.2546** | 0.0000 | 0.0000 | 0.0000 |
| Avr SubrdConj token | **0.4488** | 0.0000 | 0.0023 | 0.0000 |
| SubrdConj POS to Adj POS | **0.1985** | 0.0000 | **0.2602** | 0.0000 |
| SubrdConj POS to Noun POS | **0.2955** | 0.0000 | 0.0024 | 0.0000 |
| SubrdConj POS to Verb POS | 0.0002 | 0.0000 | 0.0000 | 0.0000 |
| SubrdConj POS to Advrb POS | 0.0002 | 0.0000 | **0.8923** | 0.0000 |
| SubrdConj POS to CordConj POS | 0.0000 | 0.0000 | **0.2257** | 0.0000 |
| Avr CordConj POS sent | 0.0000 | 0.0000 | 0.0000 | 0.0000 |
| Avr CordConj POS token | **0.2407** | 0.0000 | **0.4773** | **0.7789** |
| CordConj POS to Adj POS | **0.0208** | 0.0000 | **0.1113** | **0.2777** |
| CordConj POS to Noun POS | **0.2194** | 0.0000 | 0.0037 | 0.0056 |
| CordConj POS to Verb POS | 0.0000 | 0.0006 | 0.0000 | 0.0000 |
| CordConj POS to Advrb POS | 0.0004 | 0.0000 | **0.0220** | 0.0000 |
| CordConj POS to SubrdConj POS | 0.0000 | 0.0000 | **0.0457** | 0.0000 |
| Avr Content Words sent | 0.0017 | **0.6532** | 0.0000 | 0.0000 |
| Avr Content Words token | 0.0000 | 0.0000 | **0.4892** | **0.1905** |
| Avr Function Words token | **0.0198** | 0.0000 | 0.0000 | 0.0000 |
| Avr Function Words token | 0.0000 | 0.0000 | 0.0000 | 0.0000 |
| Content to Function Words | 0.0000 | 0.0000 | 0.0000 | **0.3664** |

Table 3: Part of speech based features.

## 4.4 Type Token Features

Aside from looking at the average or ratio-based densities of POS tags locally, syntactic complexity can also be measured via densities of collective (more than one) POS tags per sentence. Table 4 details 5 type-token ratio (TTR) based measures: *simple type-token ratio* (O'Loughlin, 1995) , *corrected type-token ratio* (Carroll, 1964), *bilogarithmic type-token ratio* (Herdan, 1960), *Uber index* (Dugast, 1978), and *simple lexical diversity*. Type token features provide a quantified measure of unique word types (ex., combination of nouns, verbs, adjectives, and adverbs) normalized by the total number of words in a segment of a language. The variations of TTR have been studied over the years to minimize the effects of sentence length when calculating the values (Herdan, 1960; Tweedie and Baayen, 1998). From the results, the Uber index and the corrected TTR metric are the only two non-significant variables from texts produced by humans and uniformly finetuned GPT-2, respectively. This may suggest fine-tuning the GPT-2 model with uniformly complex data may observe some level of uniform lexical diversity with respect to the prompt, but not for all possible metrics, and further highlights the need for the exploration of methods of generating uniform complexity in text generation.

| Feature | Human | GPT2 | GPT2-DF | GPT2-UF |
|---|---|---|---|---|
| Simple TTR | 0.0000 | 0.0000 | 0.0000 | 0.0000 |
| Correlated TTR | 0.0000 | 0.0109 | 0.0000 | **0.5532** |
| BiLogarithmic TTR | 0.0000 | 0.0000 | 0.0000 | 0.0000 |
| Uber Index | **0.6039** | 0.0000 | 0.0000 | 0.0000 |
| Lexical Diversity | 0.0000 | 0.0000 | 0.0000 | 0.0000 |

Table 4: Type token based features.

| Feature | Human | GPT2 | GPT2-DF | GPT2-UF |
|---|---|---|---|---|
| Simpl Noun variation | 0.0004 | 0.0000 | 0.0000 | 0.0000 |
| Sqrd Noun variation | 0.0000 | 0.0000 | 0.0000 | 0.0000 |
| Corr Noun variation | 0.0000 | 0.0000 | 0.0000 | 0.0000 |
| Simpl Verb variation | 0.0000 | 0.0000 | 0.0000 | 0.0000 |
| Sqrd Verb variation | 0.0000 | 0.0000 | 0.0000 | **0.6341** |
| Corr Verb variation | 0.0000 | **0.1771** | 0.0000 | 0.0053 |
| Simp Adj variation | 0.0000 | 0.0000 | 0.0000 | 0.0000 |
| Sqrd Adj variation | 0.0000 | 0.0000 | 0.0000 | 0.0000 |
| Corr Adj variation | 0.0000 | 0.0000 | 0.0000 | 0.0000 |
| Simp Adv variation | **0.1610** | 0.0000 | 0.0000 | 0.0000 |
| Sqrd Adv variation | 0.0000 | 0.0000 | 0.0000 | 0.0000 |
| Corr Adv variation | 0.0000 | 0.0000 | 0.0000 | 0.0000 |

Table 5: Lexical variation based features.

## 4.5 Lexical Variation Features

In complement to calculating ratios and averages of word-level POS complexities, lexical variation can also signal difficulty via densities of unique grammatical components (Lu, 2012). Table 5 describes 12 lexical variation-based features focusing on *simple, squared,* and *corrected versions* of unique counts of POS such as *nouns, verbs, ad-*

*jectives* and *adverbs* normalized its total in a sentence. From the results, only the unique adverb and verb variations from the human-generated, baseline GPT-2 model, and uniformly complex GPT-2 model obtained non-significance. Complementing the results previously seen in POS features, we further see the evidence of how fine-tuning may impact text continuations, especially on the usage of verbs that usually denote events and actions in a story.

## 4.6 Phrasal Features

Moving on to longer sequences of part-of-speech complexities, we also measure phrase-level linguistic features. Table 7 shows 42 phrase-based features centering on token and sentence ratios of grammatical components such as *noun phrases, verb phrases, adverbial phrases, adjectival phrases, subordinate phrases* and *prepositional phrases*. Overall, the human continuation obtained 12 phrasal-based features, which were uniform with respect to the prompt, 13 for the base GPT-2, and 7 for the finetuned GPT-2 model for general story generation, and 8 for the finetuned GPT-2 model with uniformly complex data. From this result, we posit that not performing fine-tuning of the GPT-2 model preserves the syntactic structure of the sentence at phrase level but not at finegrained word level as seen in POS features in Table 3. We also see a connection as to why Transformer-based models are often finetuned for paraphrasing tasks to trigger the required number of lexical swaps and syntactic diversity (Witteveen and Andrews, 2019; Krishna et al., 2020).

| Feature | Human | GPT2 | GPT2-DF | GPT2-UF |
|---|---|---|---|---|
| Avr Tree height sent | **0.0326** | 0.0000 | 0.0002 | 0.0000 |
| Avr Tree height token | 0.0000 | 0.0000 | 0.0000 | 0.0000 |
| Avr FTree height sent | **0.1125** | 0.0000 | 0.0000 | 0.0000 |
| Avr Ftree height token | 0.0000 | 0.0000 | 0.0000 | 0.0000 |

Table 6: Syntax tree based features.

## 4.7 Syntax Tree Features

Following the results of phrase-based features in Table 7, analyzing the difference in parse tree depth is a unique and interesting way of measuring readability as done in Schwarm and Ostendorf (2005). Table 6 describes 4 syntax tree height-based features including *average heights of regular and flattened trees* per token and sentence. From the result, only the human-generated texts as it obtained uniformity with the prompts as evidenced by features

such as *average regular tree height* and *average feature tree height* at the sentence level, while none for the GPT-2 models. This suggests that neural model-based generated texts do not conform to the one property of syntax, such as parse tree height, when generating texts, despite the models themselves having access to a significant amount of syntactic information.

| Feature | Human | GPT2 | GPT2-DF | GPT2-UF |
|---|---|---|---|---|
| Avr Noun phrs sent | **0.0280** | 0.0000 | 0.0000 | 0.0000 |
| Avr Noun phrs token | 0.0000 | 0.0121 | 0.0000 | 0.0000 |
| Noun phrs to Verb phrs | 0.0000 | 0.0000 | 0.0000 | 0.0000 |
| Noun phrs to SubClaus | 0.0000 | **0.0757** | **0.0222** | **0.8315** |
| Noun phrs to Prep phrs | 0.0047 | 0.0067 | 0.0000 | 0.0064 |
| Noun phrs to Adj phrs | 0.0000 | 0.0000 | 0.0000 | 0.0000 |
| Noun phrs to Adv phrs | 0.0000 | 0.0000 | 0.0000 | 0.0000 |
| Avr Verb phrs sent | **0.2050** | 0.0000 | 0.0008 | 0.0000 |
| Avr Verb phrs token | 0.0000 | 0.0000 | 0.0000 | 0.0000 |
| Verb phrs to Noun phrs | **0.0335** | 0.0000 | 0.0000 | 0.0000 |
| Verb phrs to SubClaus | 0.0000 | 0.0000 | 0.0000 | 0.0003 |
| Verb phrs to Prep phrs | **0.5188** | 0.0168 | 0.0000 | 0.0000 |
| Verb phrs to Adj phrs | 0.0000 | 0.0000 | 0.0000 | 0.0000 |
| Verb phrs to Adv phrs | 0.0000 | 0.0000 | 0.0000 | 0.0000 |
| Avr SubClaus sent | **0.7199** | 0.0000 | 0.0021 | **0.2506** |
| Avr SubClaus token | 0.0003 | 0.0000 | 0.0000 | 0.0000 |
| SubClaus to Noun phrs | **0.2813** | 0.0000 | 0.0000 | 0.0000 |
| SubClaus to Verb phrs | 0.0033 | **0.8824** | 0.0000 | **0.2659** |
| SubClaus to Prep phrs | **0.0169** | **0.2844** | 0.0000 | 0.0097 |
| SubClaus to Adj phrs | 0.0000 | 0.0000 | 0.0000 | 0.0000 |
| SubClaus to Adv phrs | 0.0000 | 0.0000 | 0.0000 | 0.0000 |
| Avr Prep phrs sent | **0.1307** | 0.0000 | 0.0000 | 0.0000 |
| Avr Prep phrs token | **0.2976** | **0.6734** | 0.0000 | 0.0000 |
| Prep phrs to Noun phrs | **0.1831** | **0.5031** | 0.0000 | 0.0000 |
| Prep phrs to Verb phrs | 0.0000 | 0.0000 | 0.0000 | 0.0000 |
| Prep phrs to SubClaus | 0.0000 | **0.0288** | 0.0014 | **0.0197** |
| Prep phrs to Adj phrs | 0.0000 | 0.0000 | **0.1481** | 0.0000 |
| Prep phrs to Adv phrs | 0.0000 | 0.0000 | **0.1013** | 0.0727 |
| Avr Adj phrs sent | 0.0000 | 0.0029 | **0.2439** | **0.7460** |
| Avr Adj phrs token | 0.0000 | 0.0000 | 0.0000 | 0.0000 |
| Adj phrs to Noun phrs | 0.0002 | 0.0075 | 0.0000 | 0.0006 |
| Adj phrs to Verb phrs | **0.3468** | 0.2768 | **0.4599** | **0.5966** |
| Adj phrs to SubClaus | 0.0000 | 0.0014 | 0.0000 | 0.0026 |
| Adj phrs to Prep phrs | **0.6291** | **0.7919** | 0.0000 | 0.0173 |
| Adj phrs to Adv phrs | 0.0000 | 0.0000 | 0.0000 | 0.0000 |
| Avr Adv phrs sent | 0.0000 | 0.0003 | 0.0000 | 0.0000 |
| Avr Adv phrs token | 0.0000 | **0.3127** | 0.0000 | 0.0000 |
| Adv phrs to Noun phrs | 0.0000 | **0.9194** | **0.4361** | 0.0000 |
| Adv phrs to Verb phrs | 0.0000 | 0.0000 | 0.0000 | 0.0004 |
| Adv phrs to SubClaus | 0.0000 | **0.7101** | **0.2759** | 0.0000 |
| Adv phrs to Prep phrs | 0.0000 | **0.0603** | 0.0000 | 0.0000 |
| Adv phrs to Adj phrs | 0.0000 | 0.0000 | 0.0000 | 0.0000 |

Table 7: Phrasal based features.

## 4.8 Psycholinguistic and Word Familiarity Features

We also look at psycholinguistic and word familiarity variables in reading using external wordlists such as the Age-of-Acquisition (AOA) database compiled by Kuperman et al. (2012) which contains over 30,000 English content words and the SubtlexUS by (Brysbaert and New, 2009) which is a compilation of over 74,000-word forms with frequency values extracted from 8,000 general films and series. These special databases contain words that are associated with various age levels that children are expected to learn when they reach the

stage. The works of Vajjala and Meurers (2016) and Chen and Meurers (2016) both have leveraged on these predictors in readability assessment and text familiarity. Using the LingFeat tool, we extract over 26 *token, lemma, and sentence-based normalizations of AOA and SubtlexUS variations*. Results from Table 8 show that not a single model has coincided with each other in terms of uniform features. The base GPT-2 model's continuations are all significantly different from the prompts with respect to the psycholinguistic features used, the finetuned GPT-2 model only obtained non-significance from the averages of token-based features from the SubltexUS dataset, and no uniformity is seen for the GPT-2 model finetuned with uniformly complex text. This suggests that fine-tuning for general story generation may introduce some control on the use of English content words but cannot be generalized to all types of fine-tuning.

| Feature | Human | GPT2 | GPT2-DF | GPT2-UF |
|---|---|---|---|---|
| AOA word sent | 0.0000 | 0.0000 | 0.0000 | 0.0000 |
| AOA word token | 0.0000 | 0.0000 | 0.0000 | 0.0000 |
| AOA lemma sent | 0.0000 | 0.0000 | 0.0000 | 0.0000 |
| AOA lemma token | 0.0000 | 0.0000 | 0.0000 | 0.0000 |
| AOA lemma Bird sent | 0.0000 | 0.0000 | 0.0000 | 0.0000 |
| AOA lemma Bird token | 0.0094 | 0.0000 | 0.0000 | 0.0000 |
| AOA Bristol sent | 0.0000 | 0.0042 | 0.0000 | 0.0000 |
| AOA Bristol token | 0.0000 | 0.0000 | 0.0000 | 0.0000 |
| AOA CortKhanna sent | 0.0000 | 0.0042 | 0.0000 | 0.0000 |
| AOA CortKhanna token | 0.0000 | 0.0000 | 0.0000 | 0.0000 |
| SubtlexUS sent | 0.0000 | 0.0000 | 0.0000 | 0.0000 |
| SubtlexUS token | 0.0000 | 0.0000 | **0.6334** | 0.0000 |
| SubtlexUS CD sent | 0.0002 | 0.0000 | 0.0000 | 0.0000 |
| SubtlexUS CD token | 0.0000 | 0.0000 | 0.0000 | 0.0000 |
| SubtlexUS FREQ sent | 0.0000 | 0.0000 | 0.0000 | 0.0000 |
| SubtlexUS FREQ token | 0.0000 | 0.0000 | **0.0182** | 0.0000 |
| SubtlexUS CDL sent | 0.0000 | 0.0000 | 0.0000 | 0.0000 |
| SubtlexUS CDL token | 0.0000 | 0.0000 | 0.0000 | 0.0000 |
| SubtlexUS SUBTL sent | 0.0000 | 0.0000 | 0.0000 | 0.0000 |
| SubtlexUS SUBTL token | 0.0000 | 0.0000 | **0.6334** | 0.0000 |
| SubtlexUS Lg10WF sent | 0.0000 | 0.0000 | 0.0000 | 0.0000 |
| SubtlexUS Lg10WF token | **0.8759** | 0.0000 | 0.0000 | 0.0000 |
| SubtlexUS SubLCD sent | 0.0002 | 0.0000 | 0.0000 | 0.0000 |
| SubtlexUS SubLCD token | 0.0000 | 0.0000 | 0.0000 | 0.0000 |
| SubtlexUS LgCD sent | 0.0000 | 0.0000 | 0.0000 | 0.0000 |
| SubtlexUS LgCD token | 0.0003 | 0.0000 | 0.0000 | 0.0000 |

Table 8: Psycholinguistics and word familiarity based features.

## 4.9 Discourse Features

For the last linguistic feature set investigated, we look at discourse in the form of *averages of unique and non-unique entity presence* that can affect working memory load as well as *local coherence distance measures* which captures distribution and transitions of entities in a passage (Barzilay and Lapata, 2008; Guinaudeau and Strube, 2013). Feng et al. (2009) previously applied these cognitively motivated features for assessing reading difficulty in the case of adults

with intellectual disabilities. Table 9 shows the 10 discourse-level features extracted from the prompt-continuation pairs where the majority of the features have non-uniformity for both humans and GPT-2 models. Without any observable pattern of uniformity, this finding generally suggests that the generated continuations have dissimilar levels of dependencies with respect to the prompts and vice versa.

| Feature | Human | GPT2 | GPT2-DF | GPT2-UF |
|---|---|---|---|---|
| Avr Entity sent | 0.0000 | **0.0881** | 0.0000 | 0.0000 |
| Avr Entity token | 0.0000 | 0.0000 | 0.0000 | 0.0000 |
| Avr Uniq Entity sent | 0.0000 | 0.0000 | 0.0000 | 0.0000 |
| Avr Uniq Entity token | 0.0000 | 0.0000 | 0.0000 | 0.0000 |
| Local Coherence PA | 0.0000 | 0.0000 | 0.0087 | 0.0000 |
| Local Coherence PW | 0.0000 | 0.0000 | 0.0087 | 0.0000 |
| Local Coherence PU | 0.0000 | 0.0000 | 0.0000 | 0.0000 |
| Local Coh Dist PA | 0.0000 | 0.0000 | 0.0107 | 0.0000 |
| Local Coh Dist PW | 0.0000 | 0.0000 | 0.0107 | 0.0000 |
| Local Coh Dist PU | 0.0000 | 0.0000 | 0.0000 | 0.0000 |

Table 9: Discourse based features.

## 5 Similarities Between Human and Model Generations

Aside from prompt-wise comparison, we also look at which linguistic complexity features from the GPT-2 models have some form of correlation with human continuations. This experiment aims to investigate which neural story generation models are potentially closer in producing more *human-like* text with respect to linguistic characteristics. For this, we use **Pearson correlation** to do the continuation-wise analysis and extract the top correlated features described in Table 10. From the results, all GPT-2 models obtained a minimum of six features from the psycholinguistics category referencing *Age-of-Acquisition* and *SubtlexUS databases* as top correlated features with human-generated texts. With this, we infer that even if there may be inconsistencies with respect to the prompts, the continuations from neural models for story generation have some level of similarity with human continuation against complexity features such as content word usage and word familiarity from the *SubtlexUS database*. This evidence is further strengthened by triple-occurring features across all GPT-2 models in the Table, which specifically include *incidence of Age-of-Acquisition lemmas and words* and *tokens matched from the SubtlexUS databases*.

| GPT-2 | | GPT2-DF | | GPT2-UF | |
|---|---|---|---|---|---|
| SMOG | 0.188 | Avr Syll token | 0.155 | AOA word token | 0.161 |
| Avr Syll token | 0.187 | AOA Bristol token | 0.102 | AOA lemma token | 0.158 |
| SubtlexUS Lg10WF token | 0.177 | AOA CortKhanna token | 0.102 | Adv phrs to Noun phrs | 0.130 |
| SubtlexUS CD token | 0.162 | AOA lemma token | 0.102 | Verb phrs to Noun phrs | 0.113 |
| SubtlexUS SubLCD token | 0.162 | AOA word token | 0.097 | Avr Adv POS token | 0.109 |
| SubtlexUS CDL token | 0.159 | SubtlexUS CD token | 0.093 | Avr Adv phrs token | 0.108 |
| SubtlexUS LgCD token | 0.154 | SubtlexUS SubLCD token | 0.093 | SubtlexUS SubLCD token | 0.100 |
| Avr Verb POS token | 0.120 | SubtlexUS CDL token | 0.090 | SubtlexUS CD token | 0.100 |
| AOA lemma token | 0.119 | Avr Adv phrs token | 0.080 | SubtlexUS CDL token | 0.094 |
| AOA word token | 0.116 | Adv phrs to Noun phrs | 0.079 | Avr Adj phrs token | 0.093 |

Table 10: Top correlated complexity features of zero-shot GPT-2 and finetuned GPT-2 models with human continuations. Features highlighted in light green denote double occurrences between two models, while features highlighted in dark green means occurrences across all models.

# 6 Benchmarking NLG Systems with UCTG

The evaluation of natural language generation (NLG) systems is an important process toward ensuring their usability, accessibility, and accuracy in practice. This research area has become one of the most active in the NLP community in recent years due to the rise of more complex and powerful generative models (Gehrmann et al., 2021; Caglayan et al., 2020). Synthesizing the results shown from Sections 4 and 5 in this study reporting the current inconsistencies of finetuned GPT-2 models in generating uniformly complex texts, one may reason out the potential of UCTG as a benchmark task for existing NLG systems to gather attention and encourage work from the research community. For this proposition, we highlight three major points below as insights that may be considered for building more uniformly complex text generation models.

## 6.1 Advantages of considering user background.

Despite being the main focus in readability assessment research, text complexity is a century-old research area that dates back to the 1920s (Thorndike, 1927), and is a task-agnostic multi-faceted variable that can be applied in almost all language-related experiments. This makes UCTG, unlike other automatic metrics restricted to certain tasks such as BLEU (Papineni et al., 2002) or METEOR (Banerjee and Lavie, 2005) for machine translation, likely to have more impact towards accessibility and user-dependence of NLG systems if carefully considered. However, the current methods for evaluating text complexity are wide-ranging–as evidenced by the 160 linguistic features used in this study which can potentially be used as predictors for

building complexity assessment models. Thus, we recommend a **user-centric approach** for UCTG where the linguistic predictors that will be used for evaluation are catered towards specific groups of users and their language capabilities. We see strong promising support for this direction, as evidenced in previous works where different word complexity predictors (e.g. syllable length) are correlated between subgroups of people with distinct language backgrounds such as native vs. non-native English speakers (Gooding and Tragut, 2022; Gooding et al., 2021).

## 6.2 Advantages of using language-specific readability metrics.

Current disadvantages of automatic evaluation metrics include weak correlation and reflection with human preferences as well as limitations in considering finer-grained information during assessment (ex. sentence-level processing) (Sai et al., 2021; Reiter, 2018; Reiter and Belz, 2009; Stent et al., 2005). Moreover, works on expanding NLG research towards multilingual data have also gained significant traction (Gehrmann et al., 2021; Hu et al., 2020; Ponti et al., 2020). In this regard, more and more efforts from the NLG community, such as the work of Novikova et al. (2017), for instance, support the proposal for data-independent automatic evaluation methods, especially for multilingual and cross-domain applications. While this proposition has clear benefits, there is still limited work on text complexity research on the *transferability* of linguistic complexity predictors from one language to another. And so far, previous works have emphasized the importance of still using **language-specific complexity predictors** such as in Imperial and Kochmar (2023) and Imperial et al. (2022) for

Philippine languages, and Weiss et al. (2021) for German. Thus, for applying UCTG to evaluate the uniform complexity of multilingual NLG systems, we still recommend going back to the basics, such as training models with linguistically plausible and domain-specific features as starting points.

### 6.3 Diversity in selecting the right text complexity metrics.

Finally, we emphasize that the method of benchmarking NLG systems with UCTG is not strictly prescriptive and linear as with other evaluation metrics used in text generation. The choice of selecting the appropriate combination of linguistic features to measure the complexity of machine-generated texts is **suggestive and entirely up to the researcher's judgment**. For example, one may exclusively select vocabulary-based metrics deriving from psycholinguistic features similar to the ones reported in Table 8 for language learning tasks such as question generation as done in (Jiao et al., 2023). Moreover, a researcher can also train machine learning models from an extensive set of selected linguistic features as *scorers* of text complexity as done in Lee et al. (2021). The variety of candidate linguistic features may be selected with consideration of the domain of data used for training the generative models, its target users (as discussed previously), the language intricacies and properties of the data, and the extensiveness of the evaluation of said models.

## 7 Conclusion

In this study, we introduce Uniform Complexity for Text Generation (UCTG), a new benchmark task to encourage the production of uniformly complex generations with respect to prompts. To further ground the existence of the problem, we perform the most extensive investigation of linguistic properties (160 features) analyzed from generations of GPT-2 models for the narrative generation task. We find clear inconsistencies with the models, whether finetuned or not, in terms of preserving the text complexity of generations with respect to the prompts used. However, we do note some similarities in specific linguistic variables related to word familiarity and word use between humans and GPT-2 models in how they produce continuations to a prompt. Recognizing the potential and benefits of UCTG to increase the usability and accessibility of NLG systems, we discuss three major

recommendations covering the importance of user background, language dependency, and flexibility of implementation if UCTG is to be implemented as an automatic evaluation metric or benchmark task for any arbitrary generative models from here onwards.

## Limitations

We discuss clarifying statements below regarding the scope and limitations of this study.

**On the specific use of GPT-2 models.** While the possibility of using more advanced models such as GPT-4, and ChatGPT may come to one's option for nearly all generative tasks, we emphasize that the availability of these models *do not* disregard the rich literature of narrative generation work using GPT-2. Likewise, models such as ChatGPT and GPT-4 cannot technically and freely be considered baseline models for comparison as the developers of these models provided limited transparency in their documentation and a paywall for their usage.

**On proving the research problem exists.** We emphasize that the goal of this study is to prove that the research problem, the challenge of maintaining the complexity of generated responses with respect to prompts, *exists* rather than proposing experimenting with a slew of controllability methods for LLMs without a proper in-depth grounding of the problem. Our empirical investigation makes use of a full linguistic analysis of prompt-continuation pairs from humans and neural language models. We explore methods such as fine-tuning as the standard way of calibrating LLMs. We show that this task may be complex in nature, and fine-tuning may not be a direct solution to the problem, thus an open research opportunity for researchers in the field to explore.

**On experiments exclusively with English data.** All experiments, findings, and insights in this work only apply to English, as evidenced by the language of the datasets used. Thus, our findings may not generalize if similar research derived from this work is to be done with other languages using other models, such as those trained with multilingual data.

**On varying complexities with increasing length.** Our experiments, particularly with the generation

setup as described in Section 3, focus on a specific limit of tokens (minimum 150 and maximum 300) for uniformity of analysis across models. The complexity of a text may not be directly linear when substantially longer passages are generated beyond our experimentation. We leave the fine-grained exploration of measuring complexities of text segments and chunks to future work.

**On use of variables beyond linguistic features.** Our experiments and findings specifically focus on possible inconsistencies found in texts generated by LLMs using linguistically motivated features such as vocabulary use, syntax, discourse, and part of speech. We note that there are other forms of analysis done previously using information theory concepts such as entropy and uniform information density principle on tasks such as dialogue-based data (Giulianelli and Fernández, 2021; Giulianelli et al., 2021). We currently have no proof of how these concepts relate or connect to inconsistencies associated with linguistic features but leave this to future work.

## Ethics Statement

The datasets used, WRITINGPROMPTS, NEWSELA, and ELG, are all available for researchers through the links provided, with some minor information needed to provide the purpose of use. In addition, we do not involve any human participants in this study as it is centered on text complexity which is an automatic evaluation by nature. Similar to the GPT-2 model finetuned by DeLucia et al. (2021), the GPT-2 model finetuned with uniformly written texts (NEWSELA, ELG) has been made available in Huggingface.

## Acknowledgements

We thank the anonymous reviewers for their constructive feedback and the ACs, SACs, and PCs for their appreciation of this work. We also thank Alexandra DeLucia and Ekaterina Kochmar for their valuable feedback on the initial version of this work and Mark Townsend for the assistance with configuring the experiments with the Hex GPU cloud of the Department of Computer Science at the University of Bath. JMI is supported by the UKRI Centre for Doctoral Training in Accountable, Responsible and Transparent AI (ART-AI) [EP/S023437/1] of the University of Bath, the NU Research Faculty Program (Project ID: 2021F-2T-01-MLA-CCIT), and the Study Grant Program of National University Philippines.

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

## A    Uniform Barplot Visualization per Linguistic Feature

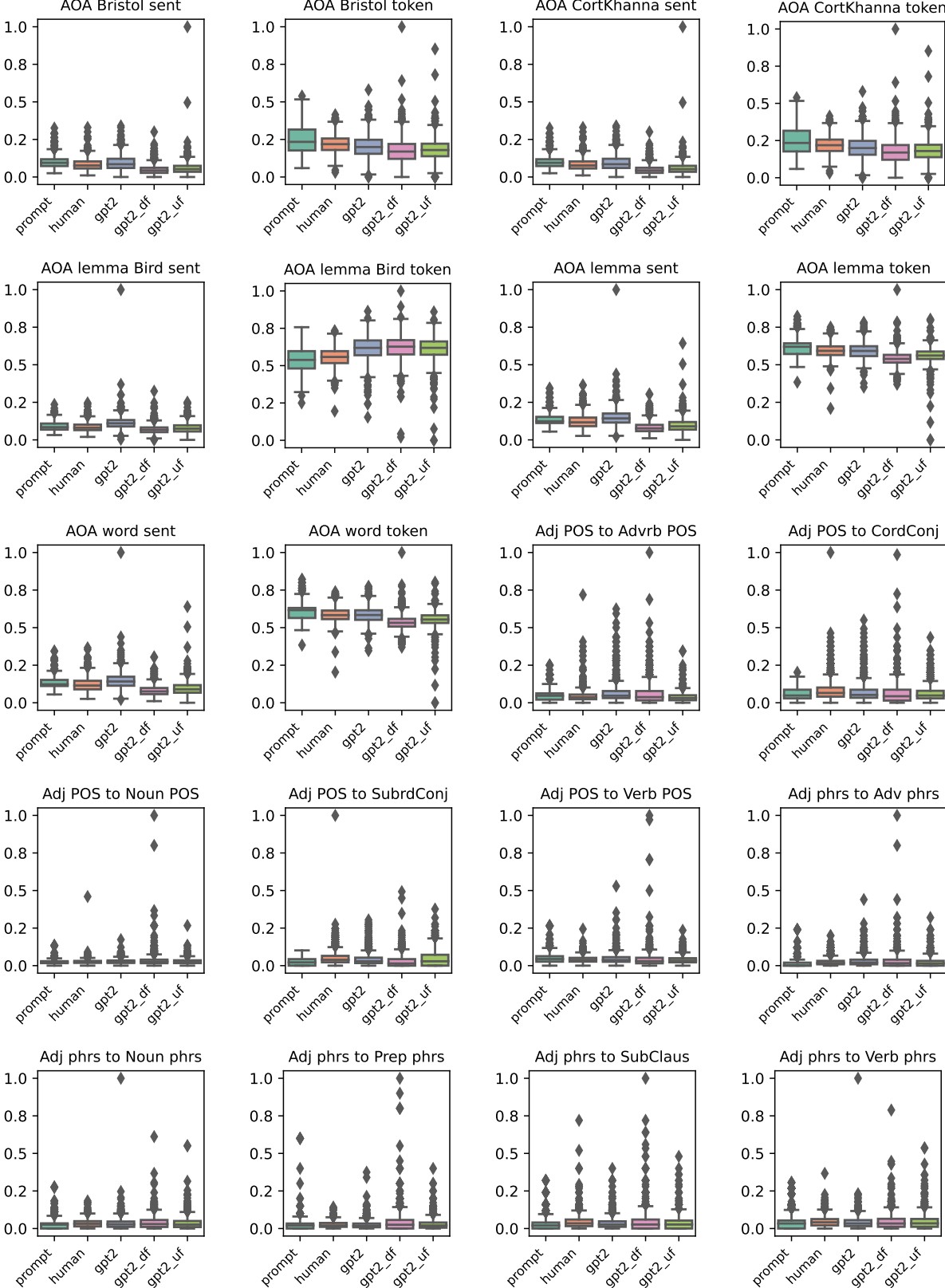

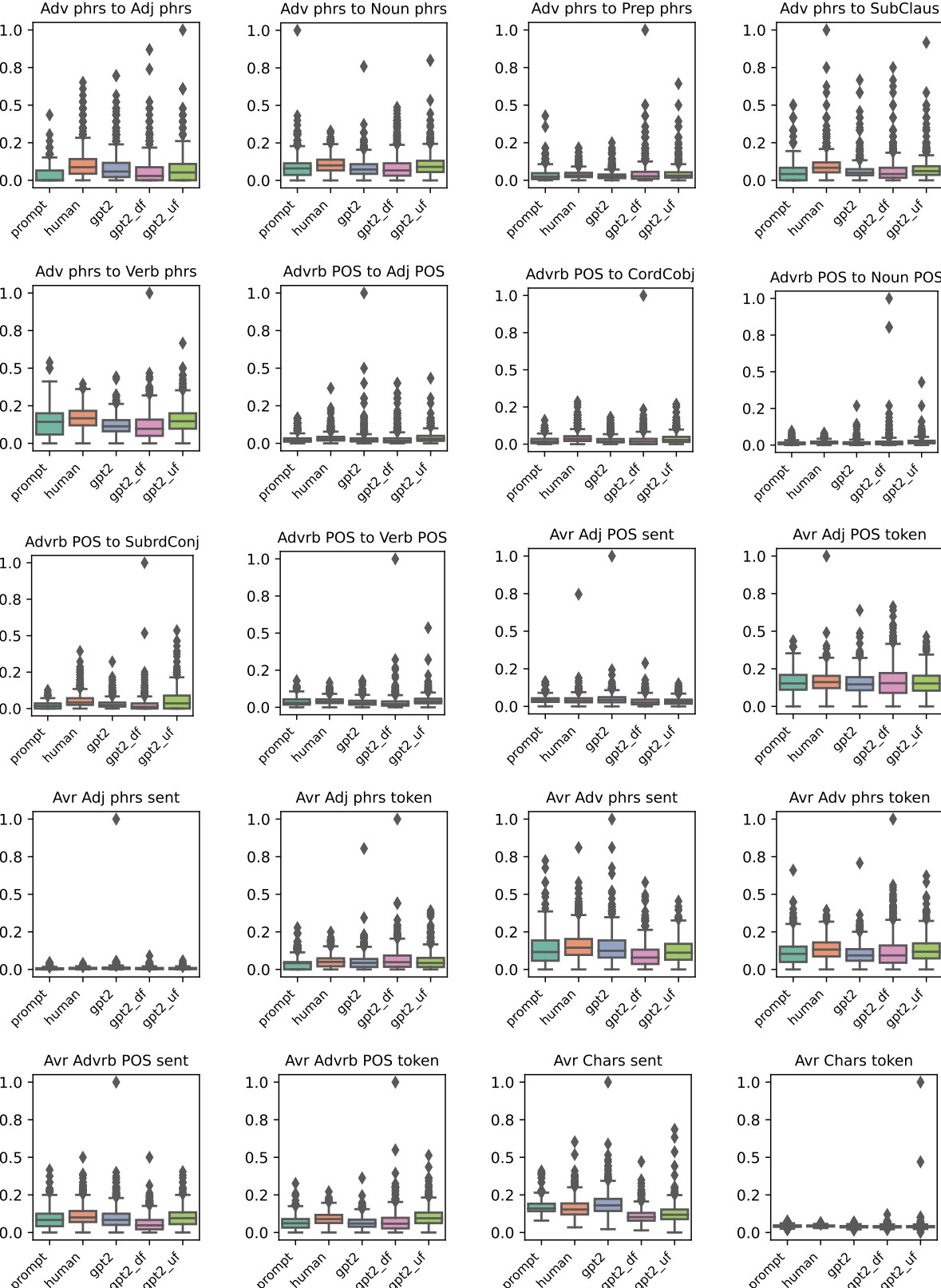

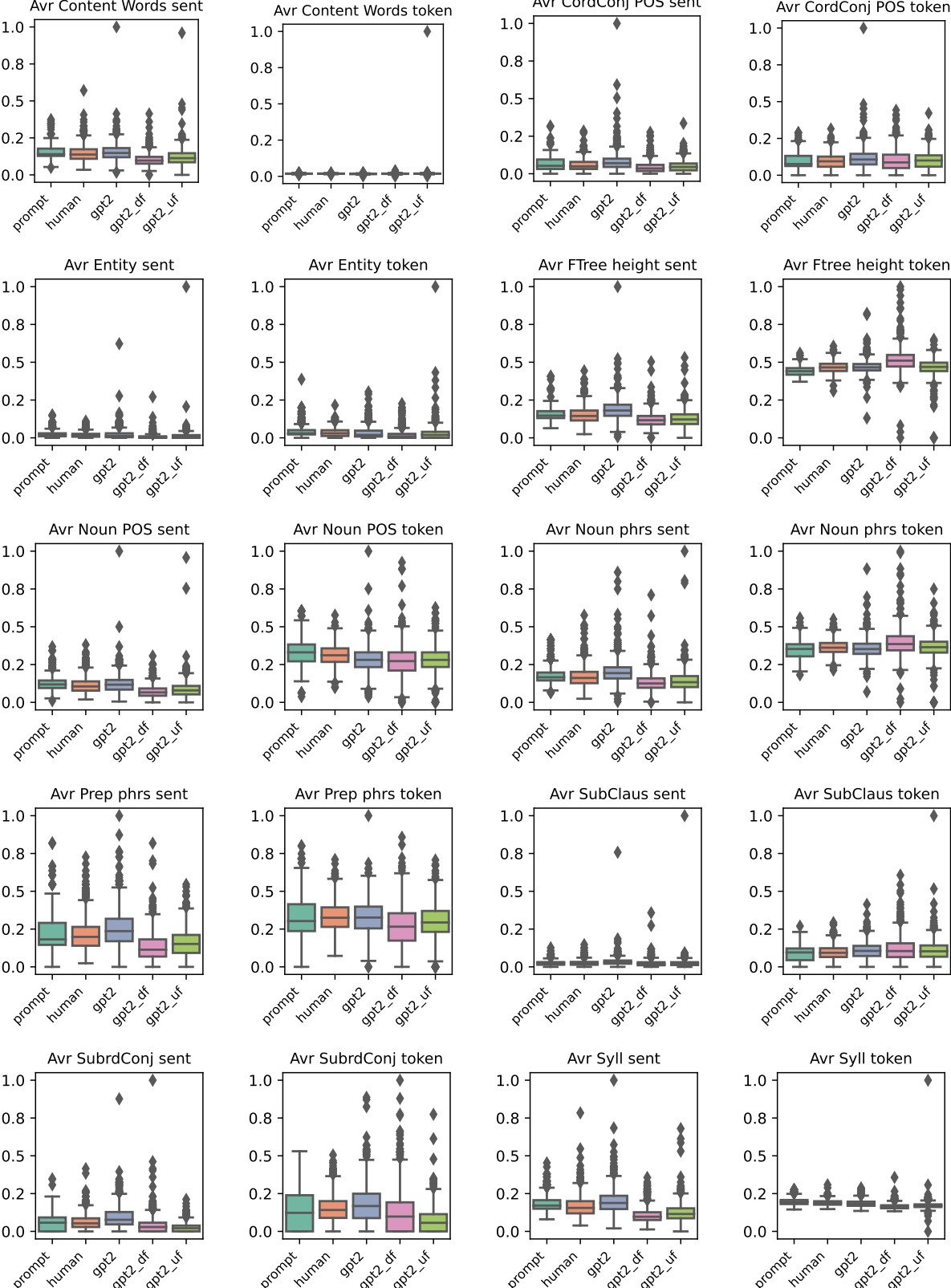

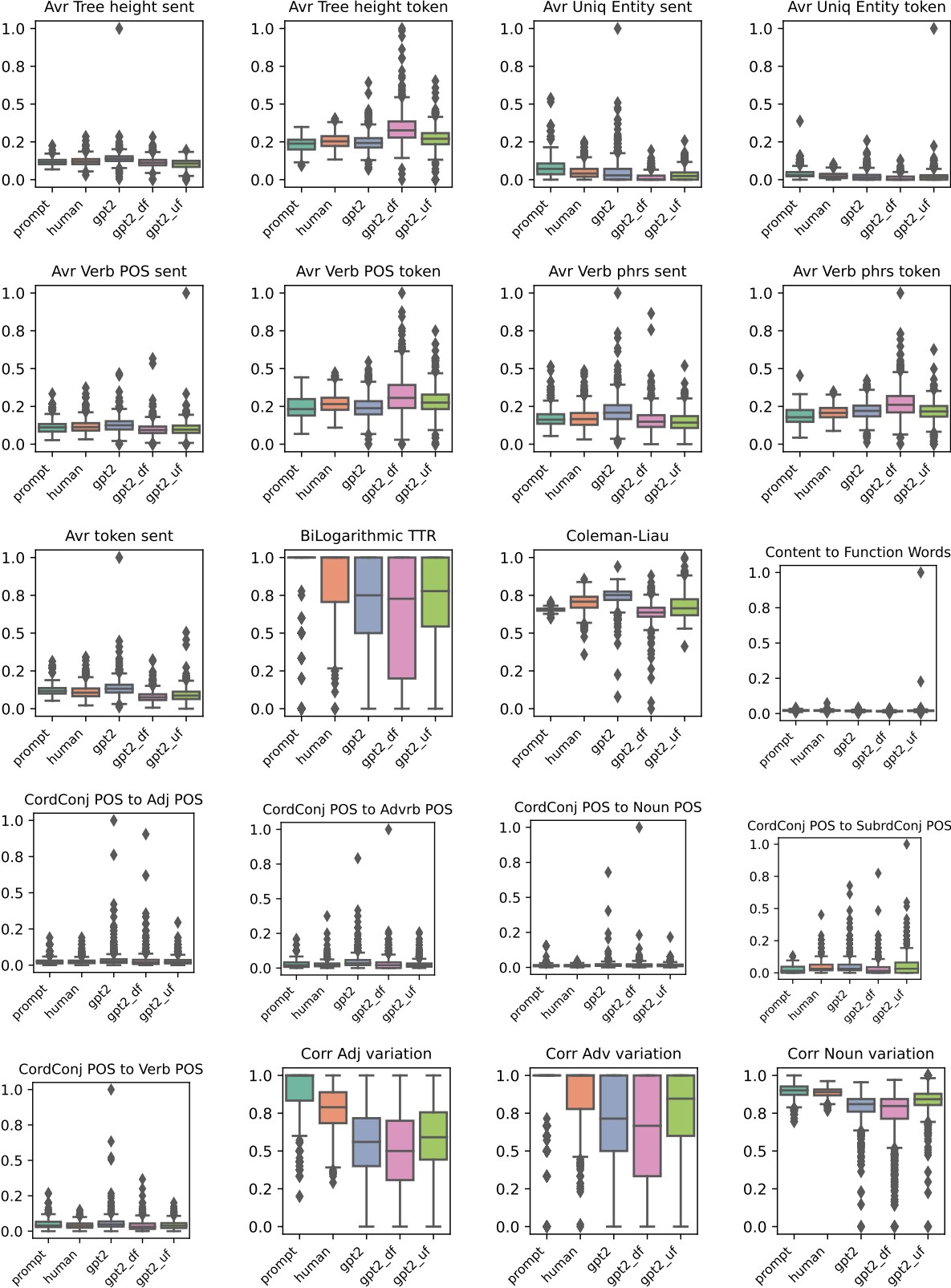

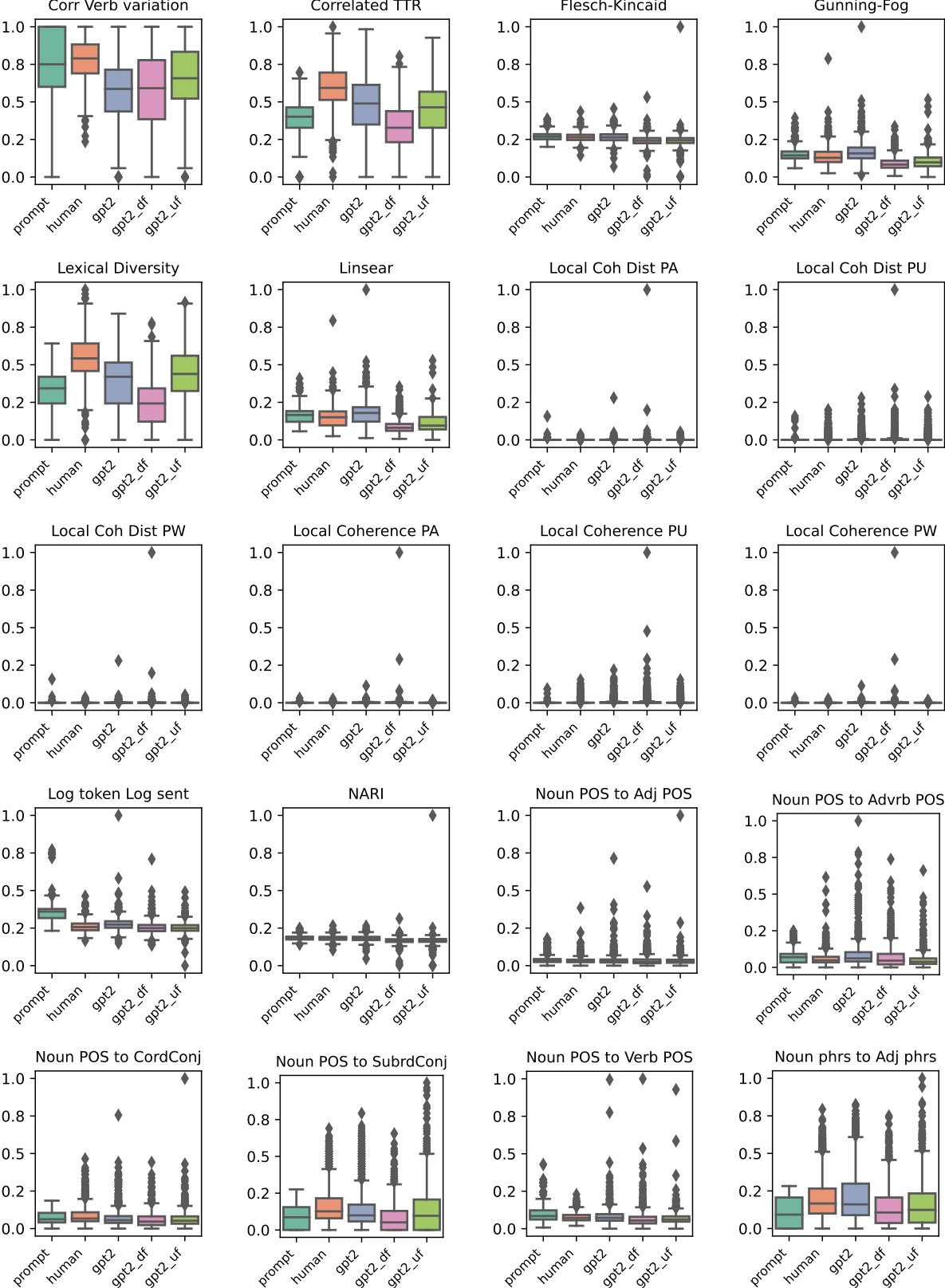

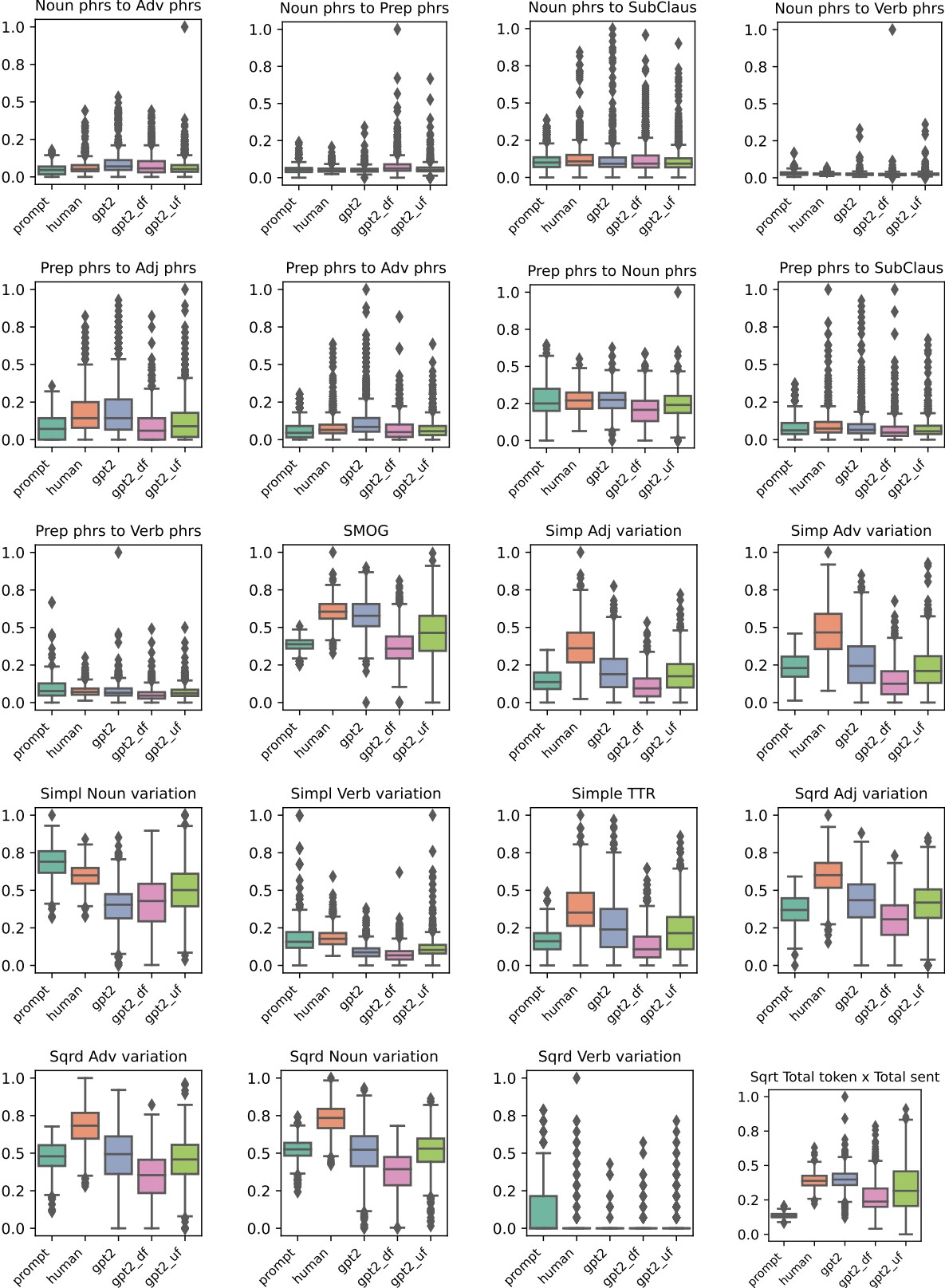

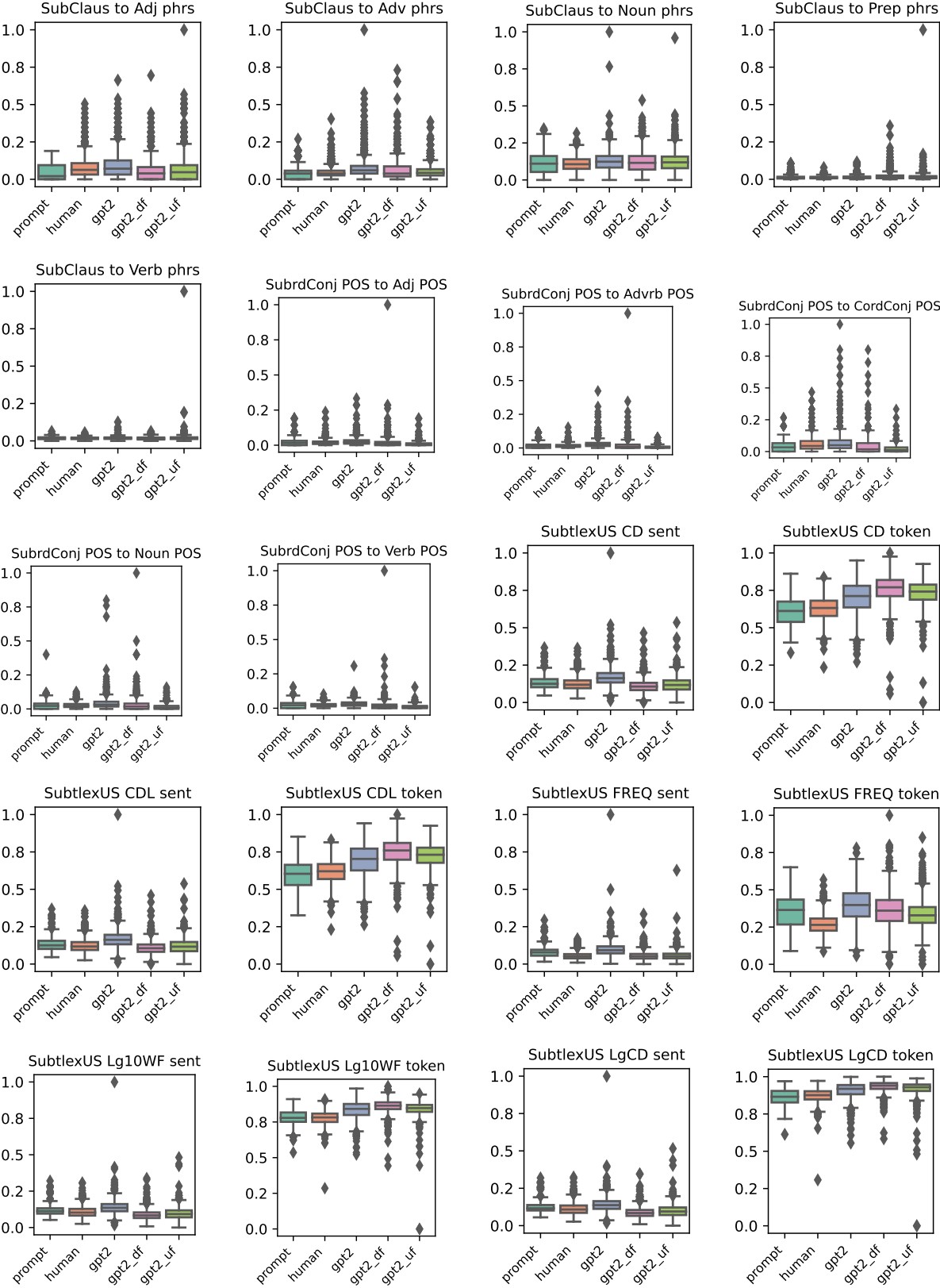

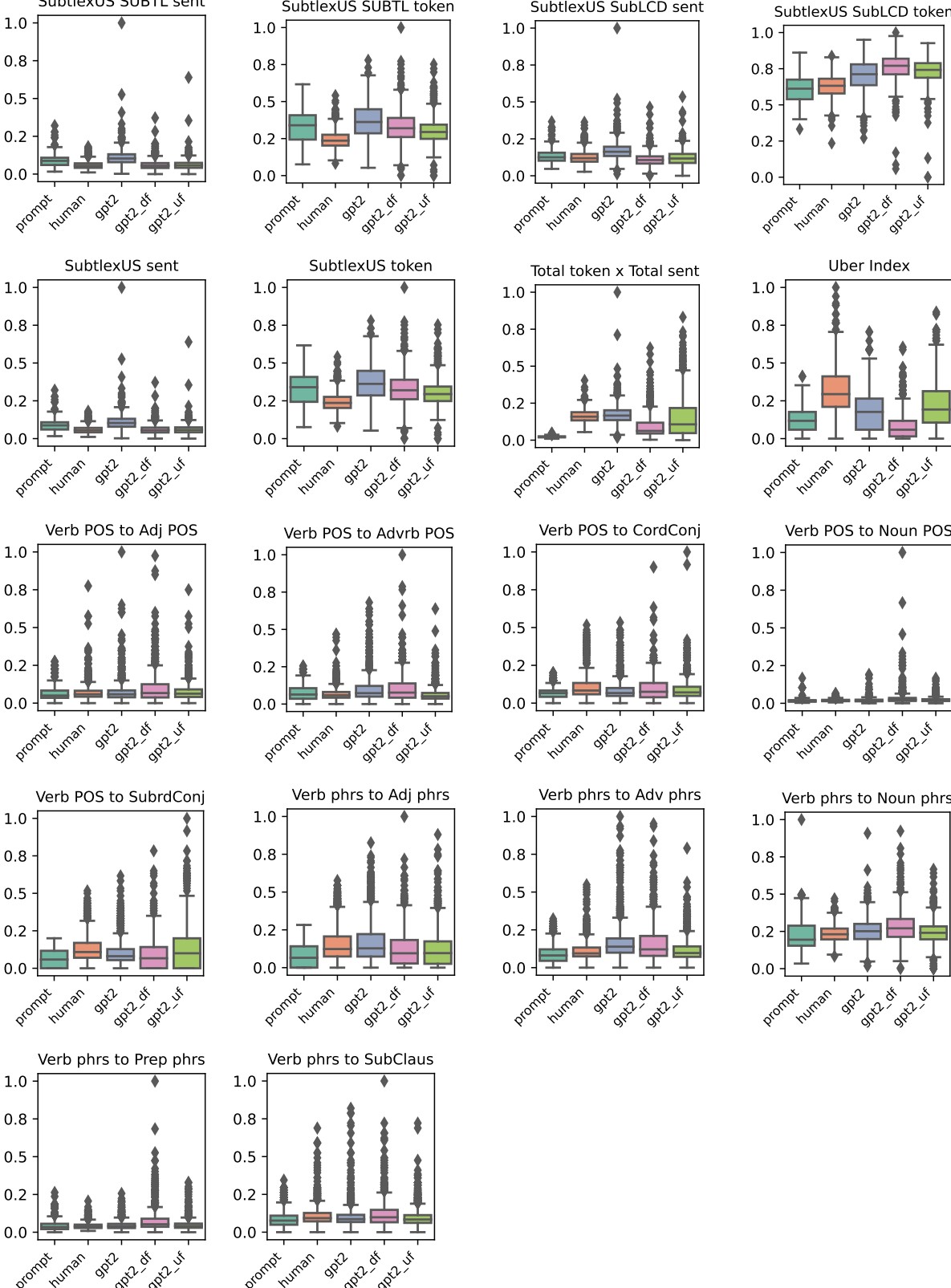