# OpenReview forum: "Uniform Complexity for Text Generation"
_EMNLP/2023/Conference — EMNLP 2023 Findings_

### Official Review · Reviewer_udN3 · 2023-07-26

**Soundness:** 4

**Excitement:**

4: Strong: This paper deepens the understanding of some phenomenon or lowers the barriers to an existing research direction.

**Paper Topic And Main Contributions:**

This paper introduces a new task, Uniform Complexity for Text Generation (UCTG), aimed at studying whether generative language models are capable of generating continuations with the same level of complexity w.r.t. the given prompts. The authors test 176 features by comparing human prompts against continuations (i) written by humans and (ii) produced by several models from the GPT-2 family. The results show that both humans and GPT-2 continuations are inconsistent w.r.t. the complexity of the given prompts. However, they show that there are some similarities in the way humans and GPT-2 generate continuations.

**Questions For The Authors:**

- A: Has any other attempt been done on testing the complexity of the text generated by LLMs? If so, it would be advisable to add a Related Work section.
- B: Line 145: can the same prompt have different continuations?
- C: Line 468-69: you state "highly-correlated features". However, in Table 10, correlation is 0.188 at max. Is that an error? Plus, are they statistically significant? Please elaborate.

**Reasons To Accept:**

- The work stems from an interesting idea, which is linguistically and cognitively motivated, introducing a novel task.
- The paper is well-written and clearly organized.
- The results are thoroughly reported.

**Reasons To Reject:**

- The assumption that a piece of text should be uniformly complex is not necessarily true. Several counter-example can be found: pedagogical texts, sub-stories in longer texts, change of complexity for some particular effects etc. Part of this point is explained and mitigated in the limitations, however, I would like to see a bit more on this.
- Only GPT models have been tested, partially limiting the generalizability of the results. However, I do believe that this can be left as a future work.
- A manual analysis on texts that received particularly high and/or low scores would have been highly appreciated.

**Reproducibility:**

5: Could easily reproduce the results.

**Reviewer Confidence:**

4: Quite sure. I tried to check the important points carefully. It's unlikely, though conceivable, that I missed something that should affect my ratings.

**Typos Grammar Style And Presentation Improvements:**

- Line 195: 0.70.
- It would be advisable to put PDF or SVG figures.
- Tables are not progressively presented in-text, e.g, Table 4 is before Table 3, 6 before 5, etc.
- Could you add in one of the captions what the bold values in the tables stand for?

---

> ### Author Rebuttal · Authors · 2023-08-28
>
> We are grateful to the reviewer for raising important points that we should consider in our study. Please find our responses to your individual questions below. We have taken note of all suggested improvements to our paper in terms of presentation, grammar, references, and content revisions. We hope that our responses are able to sufficiently address your questions, and we also hope that this may be reflected in updated scores.
>
> 1. *"The assumption that a piece of text should be uniformly complex is not necessarily true. Several counter-example can be found: pedagogical texts, sub-stories in longer texts, changes of complexity. Part of this point is explained and mitigated in the limitations, however, I would like to see a bit more on this."* - This is a very good angle of discussion and was also raised by Reviewer #1. Indeed, longer texts may be analyzed in a finer-grained way to see the change in complexity from one part to another. However, our current work is currently limited to analyzing prompt-continuation pairs as a whole to be comparable with the methodology of previous works (De Lucia, et al 2021; Fan et al 2019). We thank the reviewer for pointing this out and we can include this in our limitations section in our next revision. We believe this finer-grained analysis of longer texts would be a promising next step for future research.
>
> 2. *"Has any other attempt been done on testing the complexity of the text generated by LLMs?"* - Yes, there are a few works we cited throughout the paper such as the works of Roemmele et al (2017) and See et al (2019) on using a small set of 5-10 complexity metrics to evaluate GPT-2 model generations. In our work, we conducted a full linguistic investigation of 150+ features comparing generations from humans, base, and finetuned models.
>
> 3. *"(Line 145) Can the same prompt have different continuations?"* - Some prompts from the WritingPrompts dataset used have more than one continuation. However, we resorted to using only one prompt is to one continuation (1:1) ratio for the sake of uniformity of processing. Likewise, our analysis is approximated across all features, so it is not necessary for one prompt to have multiple continuations.
>
> 4. *"(Line 468-69) You state "highly-correlated features". However, in Table 10, correlation is 0.188 at max. Is that an error? Plus, are they statistically significant? Please elaborate."* - What we meant by “highly-correlated features” is relative to the table contents. While a correlation value of 0.188 (not a typo) is not “high” by any standards in statistics, it is the highest value in the table. Nonetheless, we understand that this has confused the reviewer and will be revising "highly-correlated features" to “top-correlated features” instead. Most of the features are statistically significant at α < 0.05. We thank the reviewer for pointing this out and will revise accordingly to improve the clarity of discussion.

---

### Official Review · Reviewer_bS2i · 2023-08-01

**Soundness:** 2

**Excitement:**

3: Ambivalent: It has merits (e.g., it reports state-of-the-art results, the idea is nice), but there are key weaknesses (e.g., it describes incremental work), and it can significantly benefit from another round of revision. However, I won't object to accepting it if my co-reviewers champion it.

**Missing References:**

(a) (158) I would recommend citing the transformers package from huggingface if this is, in fact, what you used.


**Paper Topic And Main Contributions:**

Summary: This paper explores whether GPT-2 maintains uniform text complexity between the prompt and the output in a story-generating setting. The authors quantify text complexity (roughly equivalent text difficulty) via a slew of metrics, which have been used in the prior literature on text readability and text difficulty. They find that models, generally, do not maintain complexity between a prompt and output, however that various psycholinguistic features (such as age of acquisition) are correlated between prompt and output. Overall, I think that the authors’ main question is very interesting and well-framed. I have a couple of major concerns about the methodology, which I detail below. In addition, there are some serious presentational issues with the paper. Together, these make me hesitant to recommend this paper for acceptance. That being said, I strongly encourage the authors to go through another round of revisions and resubmit – I think they are on track towards a really promising paper!

**Questions For The Authors:**

(a) (28) Is this just true for story writing or for text generation in general?

(b) (50) What’s the difference between text complexity and readability? Does it matter here?

(c) I would mention in the intro that you actually provide tests to quantify whether uniformity is observed for humans too!

(d) (171) One problem with this model is that you assume that humans do, in fact, produce uniformly complex material during generation. I think this assumption should be stated clearly. While I agree with the assumption, one problem is that the test you provide seem to indicate differently.

(e) (192) A number of other decoding strategies have emerged in recent years, including locally typical sampling (Meister et al., 2022) and eta/trunctation-sampling (Hewitt et al., 2022). I wonder if the results reported here generalize across decoding strategies.

(f) (Table 1) It’s never stated what the numbers in the table are. I assume that they are the p-value of the Welch t-test. Is this correct? Relatedly, sometimes the tables are referenced in the text but sometimes they are not. I would recommend referencing the tables consistently so readers know where to look to see results.

(g) (Section 3) It would be nice to have a list of these metrics and what they mean for handy reference. I would suggest compiling one and adding it to an appendix.

(h) (277) I am confused by this. Could you explain in greater detail or provide an example?

(i) (Table 10) It would be interesting to see the average correlation for each of the categories described by the subsections in Section 3. For example, the average correlation for discourse feature metrics, syntax tree feature metrics, etc… this would give a nice sense of variation by category, which would help understand broad trends in the data.


**Reasons To Accept:**

(a) The authors have identified an interesting research topic that complements previous work.

**Reasons To Reject:**

(a) The biggest reason to reject the paper is that the complexity metrics the authors use seem to be miscalibrated to their research question. The authors’ main intuition is that humans will maintain complexity between a text prompt and its continuation (an intuition that I share). However, the complexity metrics they use demonstrate complexity is not maintained for human responses. (That is, if I am reading the tables correctly; see (b) below.) This makes me think that their metrics are just too strong. I would encourage the authors to find metrics that do not show such differences for humans, and then explore whether results are the same for models.

(b) There are a number of presentational issues with the paper. The biggest one has to do with the tables reported in Section 3. What are the numbers? I can guess from the text that they are p-values from the statistical tests, however I don’t believe that this is indicated in the text of the paper. In addition to the p-values of the tests, it would be very helpful to see the actual raw data that the authors are collecting, along with some estimate of the error. Finally, given that the authors are reporting so many metrics, instead of the large number of tables, it would help to provide visualizations that can show the major trends in their data across metrics.

(c) Two comments about the models: First, the authors only test a GPT-2 model. Given that transformers are extremely easy to download and finetune these days, I would appreciate results from more models. Furthermore, if I am reading Section 2.2 correctly, the authors report results from a model that was fine-tuned on the test set (GPT-DF). It would be better to report model behavior on data that it hasn’t seen during training.

(d) Finally, the authors mention the possibility for this type of analysis (UCTG) to be the basis for an automatic evaluation metric. I think this paper would be much stronger if the authors followed through on this, and actually created some scripts to run the types of evaluations they report. This would greatly increase the value of this research to the community!


**Reproducibility:**

3: Could reproduce the results with some difficulty. The settings of parameters are underspecified or subjectively determined; the training/evaluation data are not widely available.

**Reviewer Confidence:**

4: Quite sure. I tried to check the important points carefully. It's unlikely, though conceivable, that I missed something that should affect my ratings.

**Typos Grammar Style And Presentation Improvements:**

(a) (55) Output is consistent → output to be consistent

(b) (89) that observes uniform → that observe uniform

---

> ### Author Rebuttal · Authors · 2023-08-28
>
> We are grateful to the reviewer for raising important and very detailed points to consider in our study, particularly with how the task can be framed, as well as possible additions to its limitations. We have taken note of all suggested improvements to our paper in terms of presentation, grammar, references, and content revisions. Please find our responses to your individual questions below. We hope that our responses are able to sufficiently address your questions, and we also hope that this may be reflected in updated scores.
>
> 1. *"The complexity metrics the authors use seem to be miscalibrated to their research question. The complexity metrics they use demonstrate complexity is not maintained for human responses. I would encourage the authors to find metrics that do not show such differences for humans, and then explore whether results are the same for models."* - This is a very good angle of discussion for the paper and we thank the reviewer for pointing this out. However, we cannot truly know what linguistic feature or metric would remain constant for humans and may or may not be for models unless we try all possible measures of text complexity in a generation experiment—thus, the whole methodology of the paper.
>
> 2. *"Given that the authors are reporting so many metrics, instead of the large number of tables, it would help to provide visualizations that can show the major trends in their data across metrics."* - Thank you for this recommendation.  Similar to Reviewer #1’s suggestion, we can provide some information on visualized direction by comparing the mean of each linguistic feature for the continuation whether they are lesser or greater than the prompt. We can do this by adding arrow lines (red for decrease and green for increase) for each column in the tables for our next revisions of the paper.
>
> 3. *"The authors report results from a model that was fine-tuned on the test set (GPT-DF). It would be better to report model behavior on data that it hasn’t seen during training."* - The data we used for finetuning GPT-DF are texts from the NEWSELA and ELG datasets, which are considered to be professionally-written. These datasets were not included in the generation procedure for Section 3, as the WritingPrompts dataset is the one used here.
>
> 4. *"The authors mention the possibility for this type of analysis (UCTG) to be the basis for an automatic evaluation metric. I think this paper would be much stronger if the authors followed through on this, and actually created some scripts to run the types of evaluations they report."* - This is a great point, and this is also the reason why we added Section 5 - Going Forward. We recommend that while researchers have the freedom to implement any linguistic metric using any tool or software, they would also have to consider factors such as taking into consideration user background and properties of the language they will work on. Our tone for this recommendation inclines more on being suggestive of the benefits rather than prescriptive.
>
> 5. *"(Line 28) Is this just true for story writing or for text generation in general?"* - We believe that consistency still holds true even if applied broadly to text generation, but more strictly followed in story generation.
>
> 6. *"(Line 50) What’s the difference between text complexity and readability? Does it matter here?"* - Text complexity and readability are often interchanged in previous works. However, we can further clarify this in our paper. We thank the reviewer for pointing this out.
>
> 7. *"(Line 192) A number of other decoding strategies have emerged in recent years, including locally typical sampling and truncation-sampling. I wonder if the results reported here generalize across decoding strategies?"* - We would like our experiments to complement previous works that have also used the same dataset as WritingPrompts and generation settings (Fan et al., 2018; Delucia et al. 2021) for uniform comparison. We can add the choice of decoding method used in the Limitations section as a reference for researchers who want to explore this particular research direction.
>
> 8. *"It’s never stated what the numbers in the table are. I assume that they are the p-value of the Welch t-test. Is this correct?"* - Yes, the values on the tables are the p-values of the Welch test with respect to the setup (columns) and linguistic feature tested (rows). We apologize for not being explicit about this in the discussion. We will revise Section 2 to improve the clarity.
>
> 9. *"(Line 277) I am confused by this. Could you explain in greater detail or provide an example?"* - Previous works have supported that texts for lower levels of readability are usually short and simple. Thus, the use of POS features such as coordinating conjunctions, prepositional phrases, subordinating clauses, etc. may signal longer texts and would lead to increased complexity.

---

### Official Review · Reviewer_kTEu · 2023-08-04

**Soundness:** 4

**Excitement:**

2: Mediocre: This paper makes marginal contributions (vs non-contemporaneous work), so I would rather not see it in the conference.

**Missing References:**

* Entropy Rate Constancy in Text. Dmitriy Genzel and Eugene Charniak. 2002
* Is Information Density Uniform in Task-Oriented Dialogues? Mario Giulianelli, Arabella Sinclair, Raquel Fernández. 2021
* Analysing Human Strategies of Information Transmission as a Function of Discourse Context. Mario Giulianelli and Raquel Fernández. 2021.


**Paper Topic And Main Contributions:**

This paper discusses the evaluation of text complexity and whether standard language models are able to maintain the linguistic complexity of the context on which they condition. The work discusses different classes of metrics that can be used to assess the complexity of text. It then performs a large analysis of models trained under different settings/objectives, and evaluates whether text complexity is significantly different in the prompt vs. continuation. They find that according to most metrics, the complexity of a prompt is often not maintained by the generation models; this is true under the different model settings tested. However, this seems to also be the case for human continuations of the same prompt.

**Questions For The Authors:**

* It’s unclear if these attributes are stable across human texts in general, not just in the prompt vs. continuation setting. Some sort of temporal analysis (i.e., over different chunks of a given text) would have been nice, both in the human and model generation settings. Like does complexity monotonically increase/decrease?
* What are the directions of change for the various metrics? Are they increasing or decreasing?
* It’s unclear what the values being reported in the tables actually are. Are they t-statistics, p-values,...?


**Reasons To Accept:**

* The paper is very clearly written and well-motivated
* Text complexity is an interesting aspect of generation that hasn’t received much attention. The authors convincingly argue that it should be taken into consideration
* The authors use a comprehensive set of complexity metrics in their analysis


**Reasons To Reject:**

* The authors don’t offer a single comprehensive complexity metric; while this might be difficult, its unrealistic to ask the NLG community to evaluate 176 different attributes of texts for every given system. Therefore, the practical implications of this paper are not obvious
* The notion of “uniform” in this work is limited to only the first and second half of a text, which is quite a coarse-grained analysis. Rather, looking at more fine-grained segments would arguably be more informative, to see if models are able to maintain complexity across their generations
* The analysis is limited to a rather small language model and its unclear if the results scale to the larger models that are more often used today for generation
* There is no discussion of related work


**Reproducibility:**

5: Could easily reproduce the results.

**Reviewer Confidence:**

4: Quite sure. I tried to check the important points carefully. It's unlikely, though conceivable, that I missed something that should affect my ratings.

---

> ### Author Rebuttal · Authors · 2023-08-28
>
> We are grateful for the reviewer’s very positive feedback and the score given to our work. Please find our responses to your individual questions below. We hope that our responses are able to sufficiently address and clarify your questions and concerns. We have taken note of all suggested improvements to our paper in terms of presentation, grammar, references, and content revisions.
>
> 1. *"The authors don’t offer a single comprehensive complexity metric. The practical implications of this paper are not obvious."* - As discussed in part of the Introduction (second page), unlike other metrics such as semantic similarity, text complexity is a characteristic that can be measured by a variety of content-based and linguistic factors covering syntactic, discourse, vocabulary, frequency, etc. Thus, compressing all of them into one complexity metric may trivialize the task. We hope that retaining the various features that makeup text complexity will help in alleviating this problem. We also reflect this in Section 5 - Going Forward, where we recommend that while researchers have the freedom to implement any linguistic metric to test for UCTG, they would also have to consider factors such as user background and properties of the language they will work on. Our tone for this recommendation inclines more on being suggestive of the benefits rather than prescriptive.
>
> 2. *"The analysis is limited to a relatively small language model, and it's unclear if the results scale to the larger models that are more often used today for generations."* - We understand the valid concern of the reviewer on the limitations of specifically using GPT-2. We would like to emphasize that our work is derived from the rich literature of story generation studies using GPT-2, as acknowledged in the Limitations section. Moreover, GPT-2 is still one of the open-source OpenAI models that the public can access freely. Thus, our contribution can be beneficial for future works studying LLMs, especially if factors such as scale, interpretability, and linguistic inconsistencies are considered.
>
> 3. *"Looking at more fine-grained segments would arguably be more informative, to see if models are able to maintain complexity across their generations. // Some sort of temporal analysis would have been nice, both in the human and model generation settings."* - We would like our experiments to complement previous works that have also used the same dataset as WritingPrompts (ex. Fan et al, 2018; Delucia et al 2021) in which the narratives are not excessively long (maximum 100-256 words). However, the reviewer raised a very good point, and while we cannot bring new results here due to time constraints. We will definitely explore this approach for our next revisions of the paper.
>
> 4. *"What are the directions of change for the various metrics? Are they increasing or decreasing?"* - This is a great recommendation. While we conducted a two-tailed t-test, which is inherently non-directional, we can provide some information of direction by comparing the mean of each feature for the continuation, whether they are lesser or greater than the prompt. We can do this by adding arrow lines (red for decrease and green for increase) for each column per feature in the tables after revision.
>
> 5. *"There is no discussion of related work"* - We explicitly did not add an RRL section because we wanted to scatter the comparison of previous works across all parts of the paper with their respective discussions. Previous works on evaluating generations of LLMs closely similar to ours have been cited in Section 1 - Introduction, Section 2 - Generation Setup, and Section 5 - Going Forward. We can include a separate RRL section for our next revision.

---

### Meta-Review · Area_Chair_q2HV · 2023-09-23

**Recommendation:** 3

**Metareview:**

Pros:
* The paper introduces a new task to study whether language models are capable of generating text with the similar level of complexity observed in the respective prompts.
* All reviewers agree that studying the text complexity in prompted models is an interesting and novel idea, and will be of interest to the community.
* The authors present  a through analysis for the proposed task with a comprehensive set of complexity metrics.

Cons:
* The authors needs to improve their motivation and the limitations of the study elaborating on 1) the assumption that "humans will maintain complexity between a text prompt and its continuation" (given that the complexity metrics tell a different story), 2) the fact that a long text could have varying level of complexities (in contrast this work focuses on prompt continuation only), and 3) whether or not a more fine grained text analysis would be more informative.
* All reviewers pointed out that the analysis in the paper will improve with larger set of models. Currently, only GPT-2 results are analysed.
* The paper could also improve with better presentation and visualisation of their results. The authors have responded to these requests and have agreed on improving these aspects.

---

### Decision · Program_Chairs · 2023-10-07

**Decision:**

Accept-Findings

**Comment:**

Pros:
* The paper introduces a new task to study whether language models are capable of generating text with the similar level of complexity observed in the respective prompts.
* All reviewers agree that studying the text complexity in prompted models is an interesting and novel idea, and will be of interest to the community.
* The authors present  a through analysis for the proposed task with a comprehensive set of complexity metrics.

Cons:
* The authors needs to improve their motivation and the limitations of the study elaborating on 1) the assumption that "humans will maintain complexity between a text prompt and its continuation" (given that the complexity metrics tell a different story), 2) the fact that a long text could have varying level of complexities (in contrast this work focuses on prompt continuation only), and 3) whether or not a more fine grained text analysis would be more informative.
* All reviewers pointed out that the analysis in the paper will improve with larger set of models. Currently, only GPT-2 results are analysed.
* The paper could also improve with better presentation and visualisation of their results. The authors have responded to these requests and have agreed on improving these aspects.